# The Role of L-Carnitine in Kidney Disease and Related Metabolic Dysfunctions

**Tim Ulinski [1],\*, Maria Cirulli [2] and Mohamed Ashraf Virmani [2],\***

[1] MARHEA Reference Center, Pediatric Nephrology Department, Trousseau Hospital, AP-HP. Sorbonne University, 75231 Paris, France

[2] Alfasigma, 3528 BG Utrecht, The Netherlands

\* Correspondence: tim.ulinski@aphp.fr (T.U.); ashraf.virmani@alfasigma.com (M.A.V.)

**Abstract:** Kidney disease is associated with a wide variety of metabolic abnormalities that accompany the uremic state and the state of dialysis dependence. These include altered L-carnitine homeostasis, mitochondrial dysfunctions, and abnormalities in fatty acid metabolism. L-carnitine is essential for fatty acid metabolism and proper mitochondrial function. Deficiency in kidney disease and dialysis is caused by a reduction in endogenous renal synthesis, impaired fatty acid metabolism, a lower intake due to dietary restrictions, and nonselective clearance by the dialysis procedure. Free carnitine levels <40 μmol/L in dialysis patients can lead to dialysis-related complications, such as anemia that is hyporesponsive to erythropoietin therapy, intradialytic hypotension, cardiovascular disease, and skeletal muscle dysfunction manifested as muscle weakness and fatigue. L-carnitine deficiency is also seen in acute kidney injury (AKI) resulting from trauma and/or ischemia, drugs such as cisplatin, and from infections such as covid. A persistent state of L-carnitine deficiency can further damage kidneys and lead to multi-organ failure. Carnitine supplementation has been shown to be safe and effective in improving kidney disease-related complications resulting from drug-induced toxicity, trauma, ischemic injury, infection, and dialysis, by replenishing adequate carnitine levels and rebalancing carnitine homeostasis. In this review, we will examine the protective role of L-carnitine in reducing cellular oxidative damage and maintaining mitochondrial function together with the clinical evidence for its potential use in the management of kidney disease.

**Keywords:** acute kidney disease; L-carnitine; mitochondria; anemia; erythropoietin; intradialytic hypotension; insulin resistance; cardiac function; kidney injury; dialysis

## 1. Pathophysiology of Kidney Disease and Relationship between Carnitine Metabolism and Kidney Function

*1.1. Metabolic Alterations in Kidney Disease and Potential Protective Role of L-Carnitine*

Chronic kidney disease is associated not only with a wide variety of metabolic abnormalities that accompany the uremic state but also with specific changes related to the dialysis procedure. These include altered L-carnitine homeostasis, mitochondrial dysfunction, and abnormalities in fatty acid metabolism. Carnitine deficiency is common in kidney disease and dialysis. L-carnitine has been shown to be an effective adjunctive treatment for anemia, intradialytic hypotension, hyperlipidemia, and muscle weakness.

L-carnitine is an amino acid derivative naturally produced by the body and obtained from the diet, especially from red meat. Its primary function in cells is to transport long-chain fatty acids across the inner mitochondrial membrane for β-oxidation and generation of ATP energy [1,2]. L-carnitine also plays a role in transporting potentially toxic acyl molecules out of the cells and in balancing the coenzyme A (CoA) ratio within mitochondria, acting as an indirect antioxidant. Therefore, it protects cellular membranes and prevents fatty acid accumulation. L-carnitine also controls the levels of β-oxidation and the acetyl CoA/CoA ratio, which are involved in modulating ketogenesis and glucogenesis.

Kidneys are the main organs responsible for the regulation of body fluids. They balance the volume, pH, and osmolality of the extracellular fluid and regulate the amount of sodium and water excreted. The kidneys are also specifically involved in regulating L-carnitine levels by controlling the excretion and reabsorption of L-carnitine as well as the endogenous synthesis of L-carnitine (Figure 1). The carnitine pool results from the combination of intestinal absorption, endogenous synthesis, and high tubular reabsorption [3].

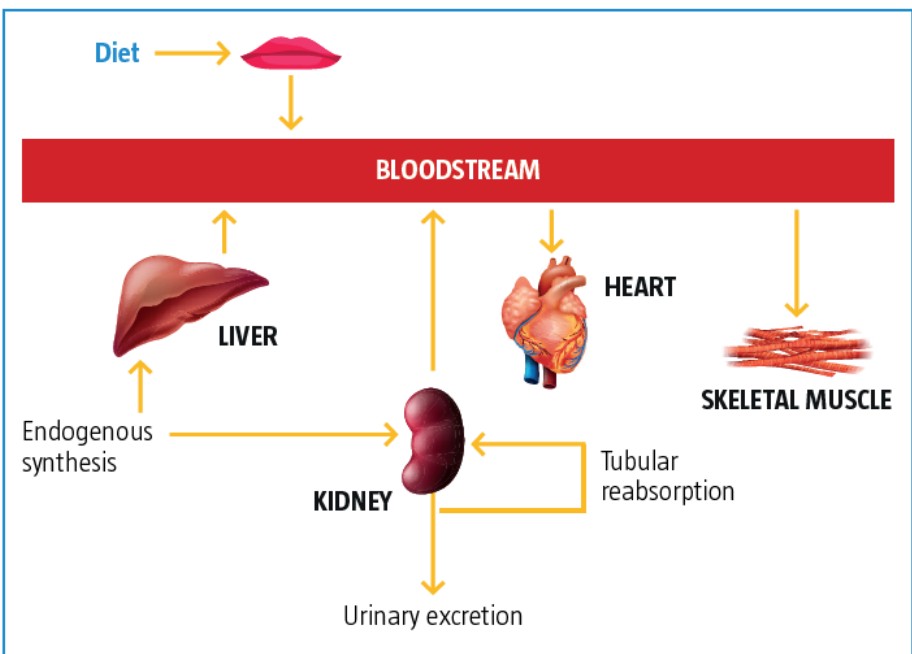

**Figure 1.** Regulation of L-carnitine levels in the blood by the diet and role of the kidney, liver, and muscle (adapted from Evans 2003 [3]). The kidney plays a very important role in maintaining blood L-carnitine levels by synthesis, excretion, and re-absorption.

The majority of L-carnitine (90–99%) filtered in the kidney is reabsorbed in the distal parts of the nephron until saturation is reached. The renal threshold for L-carnitine excretion is around 50 μmol/L. The kidneys are very efficient in maintaining normal levels of plasma L-carnitine by modulating urinary L-carnitine excretion depending on the intake from the diet [4].

At the onset of kidney disease, the glomerular filtration rate is reduced, and due to tubular dysfunction, a lower proportion of L-carnitine is reabsorbed, and the mechanism of acylcarnitine elimination is less efficient than with normal kidney function [5,6]. Initially, carnitine levels are higher, but as kidney disease progresses, more acylcarnitine is formed in the body, especially in the muscles and kidneys, while its excretion is reduced. This results in an increase in acylcarnitine in the cells and in the blood, which can lead to cellular toxicity by altering cellular and mitochondrial functions [7]. A buildup of acylcarnitine, usually due to defective β-oxidation, can increase the level of unmetabolized long-chain fatty acids (LCFA) within the mitochondria, which exert a detrimental effect on cellular membranes and proteins possibly due to a detergent-like action on the membranes [8].

As more acylcarnitine is formed, L-carnitine is decreased resulting in a free carnitine level <40 μmol/L or an acylcarnitine/free carnitine ratio of more than 0.4 in the blood, which is a sign of L-carnitine deficiency. Numerous studies investigating these changes have established that a deteriorating renal function is associated with decreased carnitine clearance and impairment of normal excretion of acylcarnitine [9,10].

L-carnitine depletion in the body may lead to frequent complications, such as anemia hyporesponsive to erythropoietin, intradialytic hypotension, muscle weakness, and cardiac arrhythmias. L-carnitine treatment has been shown to be beneficial in these dialysis-related complications [4,5,11].

*1.2. Role of the Mitochondria in Kidney Disease*

Mitochondrial dysfunction has been implicated in the pathogenesis of many diseases including kidney disease. Altered mitochondrial function leads to a reduction of ATP, an increase in ROS, and an increase in acylcarnitine, which can damage cells leading to a negative impact on kidney function in acute and chronic kidney disease states [12–17].

The preferred 'fuel' for respiration in the kidney cortex are short- and long-chain fatty acids, endogenous lipids, ketone bodies, lactate, and some amino acids (Figure 2) [18]. Chronic hyperinsulinemia and insulin resistance lead to increased degradation of triglycerides in the adipocytes as well as reduced uptake of circulating fatty acids, causing elevated serum levels of non-esterified fatty acids. These elevated levels lead to the ectopic accumulation of lipids in organs outside of the lipid tissue, including kidneys. The excessive accumulation of lipids results in cellular damage known as lipotoxicity (Figure 3). [19–21]. Fatty acids accumulating in the mitochondrial matrix are vulnerable to lipid peroxidation, which can have lipotoxic effects on DNA, RNA, and proteins that affect the mitochondrial machinery and lead to mitochondrial dysfunction as well as cellular damage [22–24].

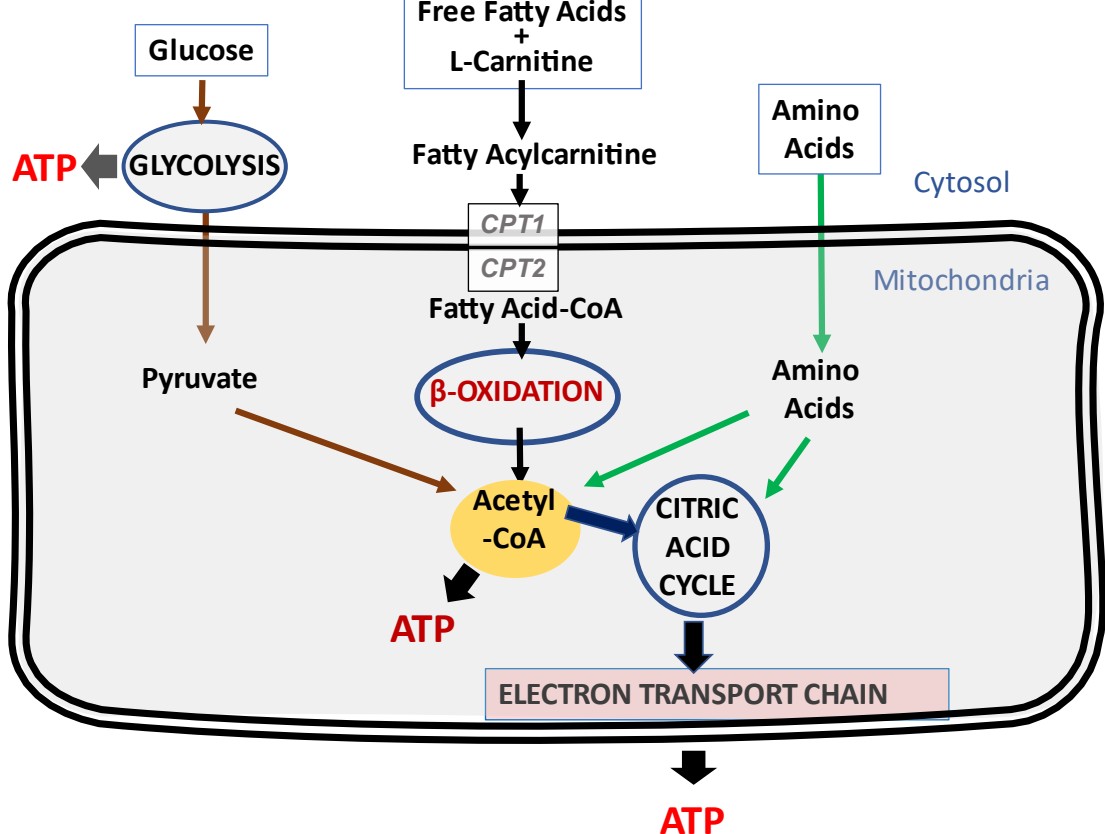

**Figure 2.** The metabolism of carbohydrates, fats, and proteins to produce ATP energy (adapted from Virmani & Cirulli 2022 [2]). Carbohydrates such as glucose can be metabolized in the cytoplasm to produce ATP whereas fatty acids need to enter the mitochondria to be broken down by β-oxidation and metabolized. There are specific enzymes, such as carnitine palmitoyl transferase 1 (CPT1) and 2 (CPT2) on the outer and inner mitochondrial membrane, respectively, to facilitate the transfer of the fatty acid combined with the L-carnitine (fatty acylcarnitine).

In AKI and diabetic nephropathy, β-oxidation in the mitochondria is decreased and the formation of lipid droplets inside the cell is increased, resulting in diminished ATP production [26–29].

Mitochondrial dysfunction also plays a key role in the pathogenesis of diabetic nephropathy, which occurs in 40% of patients with diabetes. A recent study showed that in diabetic nephropathy there is downregulation of the antioxidant superoxide dismu-

tase 2 (SOD2), whose function is to prevent the excess buildup of mitochondrial reactive oxygen species (mtROS) [30]. The increase in reactive oxygen species (ROS) can damage mitochondrial membranes and proteins, compromising mitochondrial function [31–35].

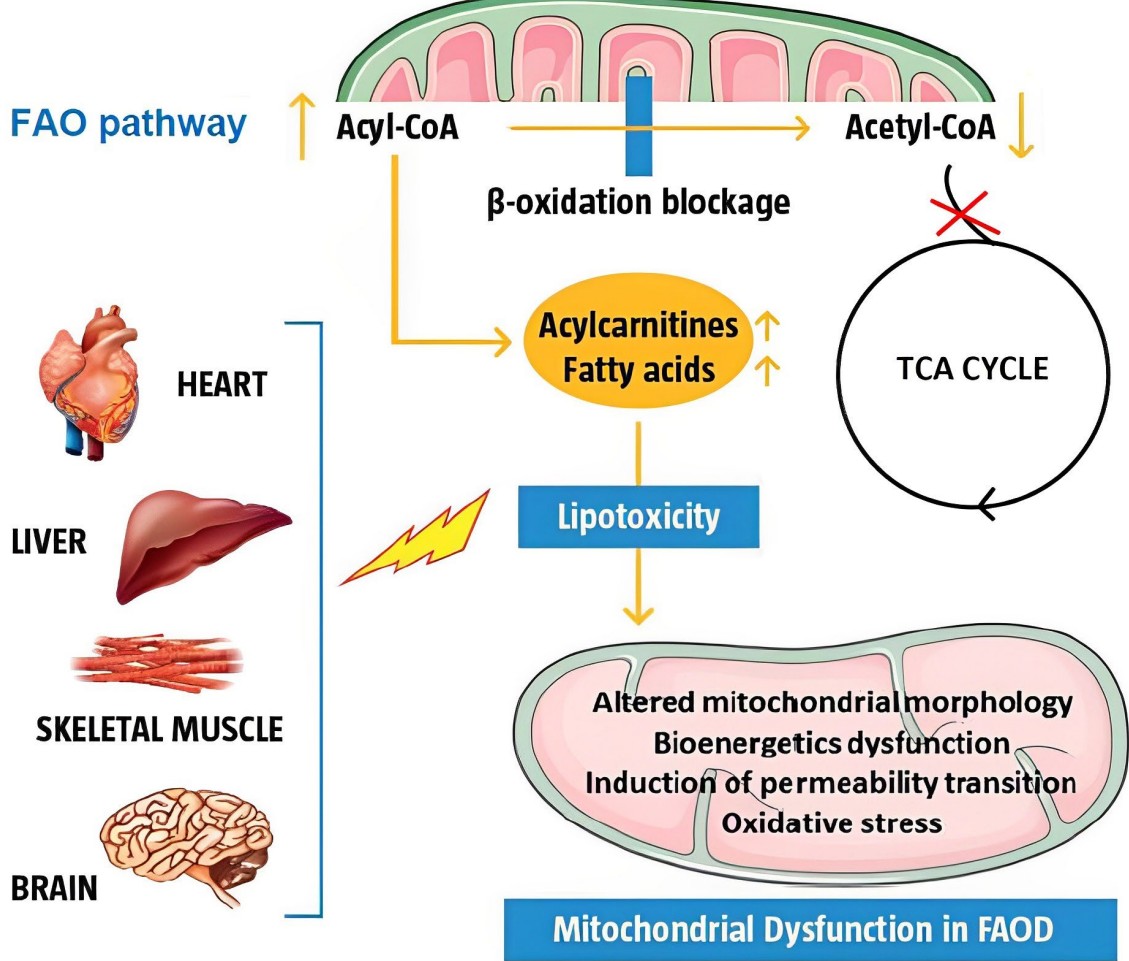

**Figure 3.** Mitochondrial dysfunction caused by fatty acids and acylcarnitines buildup due to β-oxidation blockade (adapted from Wajner & Amaral, 2016 [25]). Factors such as fatty acid oxidation disorders (FAOD) that cause inhibition of fatty acid β-oxidation in the mitochondria lead to the accumulation of the long chain acyls groups, which trigger lipotoxicity and negatively affect mitochondrial function.

## 2. Diet in Chronic Kidney Disease

A proper diet plays an important role in the management of chronic kidney disease. The diet is patient-dependent and must be personalized. There are enormous differences to consider between children and adults. Calorie and protein requirements in children are higher than in adults because they are still in the growth phase, and, therefore, protein intake must be adapted to their specific needs. This is especially true during adolescence or when they are on growth hormone therapy.

A diet low in potassium, protein, and salt is often recommended because an increased intake of these substances can lead to:

- Hyperkalemia with a risk of arrhythmia;
- An increase in plasma urea leading to neurotoxicity and cardiac toxicity;
- An increase in phosphate, which can damage the arterial structure and thereby lead to an increase of the left ventricular volume due to the increased afterload;
- Hypervolemia with a risk of arterial hypertension and pulmonary edema.

Protein in the diet can cause an increase in urea, especially as kidney function decreases [36,37]. The kidney adapts to protein intake by further increasing the glomerular filtration, leading to higher proteinuria through hyperfiltration. This mechanism is a physiological reaction to higher protein intake if the functional reserve of the kidneys is intact and helps to maintain low urea levels. However, in the case of the reduced functional nephron number, it accelerates the decline in kidney function. A decrease in protein intake reduces potentially neurotoxic urea levels and preserves the kidney from hyperfiltration. Hyperfiltration by the remaining glomeruli results in proteinuria and interstitial fibrosis. In addition, high salt intake in the diet can lead to arterial hypertension and arteriosclerosis [38–40]. Although these diet restrictions are necessary, they can often lead to insufficient levels of micronutrients that are needed for maintaining normal cellular metabolism, including mitochondrial function. Mitochondria together with cellular and body control systems is a key player in energy metabolism for the whole body, including kidneys. The buildup of toxic metabolites due to mitochondrial dysfunction can reduce metabolic flexibility. Sufficient micronutrient intake, including B vitamins, cysteine, methionine, arginine, CoQ10, and L-carnitine from the diet, is crucial in maintaining metabolic flexibility and energy balance.

A low-protein diet with a reduced intake of red meat, which is the primary source of L-carnitine, together with low L-carnitine reabsorption due to kidney malfunctioning, can lead to a vicious cycle of carnitine deficiency. This results in altered carnitine homeostasis and an increase in potentially toxic long-chain fatty acid acyl residues since one of the L-carnitine roles is to bind to acyl residues and shuttle them out of the mitochondria and the cells. Persistent L-carnitine and micronutrient deficiency contribute to metabolic dysfunctions and disturbed energy metabolism. This, in turn, leads to metabolic inflexibility, cellular damage, and, ultimately, a disease [2].

Given the multiple roles of carnitine, we can hypothesize that certain pathological situations may benefit from higher L-carnitine plasma levels. Since the uremic state can damage the erythrocyte membrane due to uremic toxins and energy metabolism is altered by kidney disease and diet, it is difficult to determine optimal L-carnitine plasma levels. Patients with stage 5 chronic kidney disease (CKD) face a variety of abnormalities with complex interactions between them and may benefit from L-carnitine levels above the normal range.

In addition, intradialytic hypotension episodes and muscle cramps in hemodialysis (HD) patients are extremely multifactorial. For example, excess dehydration during dialysis, left ventricular insufficiency, and rapid changes in sodium plasma levels can result in hypotensive episodes during the HD session [41,42]. This can make it difficult to distinguish the individual effects of each condition and, thereby, demonstrate the beneficial effects of L-carnitine supplementation because carnitine acts not only on muscle cramps but also on cardiac function and hypotensive episodes [43,44].

We can speculate that higher than normal plasma carnitine levels may further improve the complex situation in dialysis-dependent kidney disease. Studies are needed to investigate the full potential of the optimal L-carnitine dosage in patients with complex situations, such as dialysis dependency, acute kidney injury, or multi-organ failure.

## 3. L-Carnitine in Acute Kidney Injury

AKI in adults and children is associated with conditions such as sepsis, multi-organ failure, nephrotoxins, congenital heart disease, malignancies, primary kidney disease, hypotension shock, hypoxemia, and renal ischemia. These probably contribute to the increased mortality in AKI. Several authors found that L-carnitine treatment can mitigate the negative effects of acute kidney injury in both children and adults.

In children receiving continuous kidney replacement therapy (CKRT) for AKI, intravenous L-carnitine, added to total parenteral nutrition (TPN) at a dose of 20 mg/kg/day, improved myocardial strain [45]. Another study showed that L-carnitine treatment of

50 mg/kg/d L-carnitine per day added to antibiotic regimens decreased renal scarring in children with acute pyelonephritis [46].

In adults undergoing prolonged kidney replacement therapy for AKI the plasma carnitine levels can diminish, causing metabolic disturbances and potential neurological symptoms. A recent study in patients receiving long-term tube feeding and continuous renal replacement therapy (CRRT) for more than 1 week suggested that L-carnitine supplementation at a dosage of 0.5 to 1 g/day may be beneficial in reducing neurological symptoms [47].

In patients undergoing percutaneous coronary intervention (PCI), oral L-carnitine 1 g 3 times a day, 24 h before the procedure and 2 g after PCI lowered plasma neutrophil gelatinase-associated lipocalin (NGAL) concentration, a marker for kidney damage following contrast medium administration [48].

## 4. Acute Kidney Injury from Infection and Drugs

Severe infection is a frequent cause of AKI, especially in intensive care patients. Kidney injury in infections such as COVID-19 is multifactorial, with cardiovascular comorbidity and predisposing factors, such as sepsis, hypovolemia, and nephrotoxins as important contributors. The cytotoxic effect of sepsis is probably related to inflammation and reduced microcirculation [49].

Drug-induced nephrotoxicity from antimicrobial and other drugs such as cisplatin can also contribute to AKI. Oxidative stress seems to play a considerable role in cisplatin-induced nephrotoxicity [50]. An inadequate diet low in proteins and/or calories with subsequent protein catabolism puts patients at risk for increased cisplatin toxicity. Cisplatin-induced toxicity can also lead to tubular dysfunction resulting in an increase in renal excretion and a decrease in the absorption of L-carnitine. This can cause carnitine deficiency, which could pose an additional risk factor [51]. Several preclinical studies have shown that supplementation with L-carnitine can attenuate cisplatin-induced renal toxicity [50,52–54].

One of the factors underlying kidney injury could be increased inflammation caused by infection or kidney damage. An important factor underlying inflammation is oxidative stress and the excessive release of cytokines. Studies have shown that L-carnitine can reduce levels of inflammatory mediators, such as C-reactive protein (CRP), tumor necrosis factor-$\alpha$ (TNF-$\alpha$), and interleukin-6 (IL-6) [55].

## 5. Dialysis Related Complications

Hemodialysis patients can suffer from pathological conditions partially related to the dialysis procedure, such as anemia hyporesponsive to erythropoietin (EPO), intradialytic hypotension, muscle weakness, and reduced exercise capacity [56,57].

Chronic kidney disease can also cause secondary diseases such as anemia, which often requires erythropoietin and iron supplementation to maintain a target hemoglobin level between 11 and 13 g/dL. Hemodialysis can improve anemia through the reduction of uremic toxins.

Repeated blood loss during HD sessions can also worsen anemia. Loss of L-carnitine through dialytic membranes occurs in maintenance hemodialysis, resulting in potential carnitine depletion and relative increments of esterified carnitine forms. Several studies have shown that L-carnitine supplementation has clinical benefits, such as enhanced response to erythropoietin as well as improvement in exercise tolerance, intradialytic symptoms, hyperparathyroidism, insulin resistance, inflammatory and oxidant status, protein balance, lipid profile, cardiac function, and quality of life (Figure 4) [43,44,58,59].

Nephrologists have long been aware of the consequences associated with these conditions and their impact on the overall quality of life. A multifactorial approach is essential in reducing the incidence and morbidity for the optimal management of HD patients. L-carnitine treatment plays an important role in the management of these conditions and in improving the quality of life.

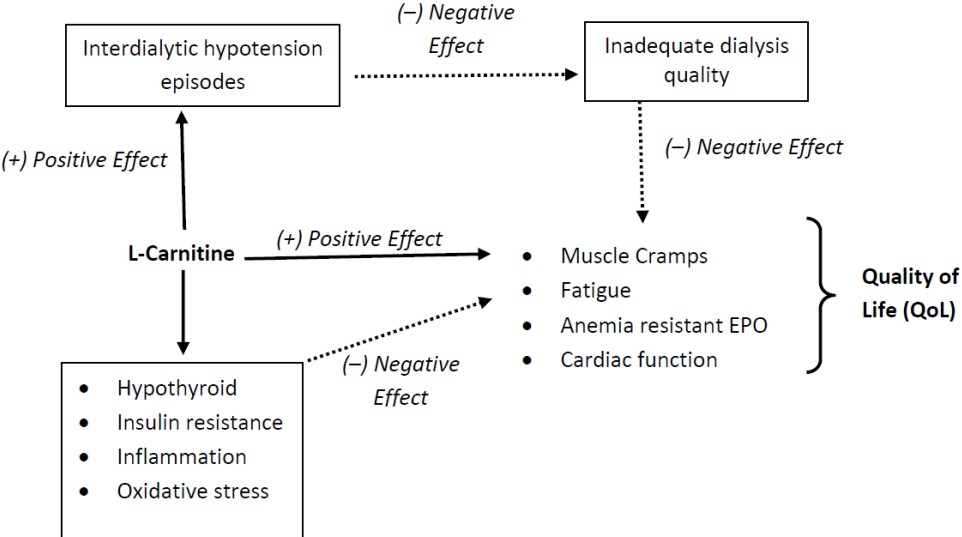

**Figure 4.** Multiple metabolic actions of L-carnitine on quality of life (QoL). L-carnitine has numerous actions that produce positive effects on blood pressure, anemia, inflammation, and oxidative stress. L-carnitine also has a positive effect on thyroid, muscle, and cardiac function. Together, these actions can result in an improved QoL for patients in dialysis.

### 5.1. L-Carnitine in Anemia and Response to Erythropoietin

Anemia is a common and important complication among patients with chronic kidney disease (CKD). Untreated anemia places patients at risk for cardiovascular events, more rapid progression of renal disease, significantly decreased quality of life, and death [60,61].

In dialysis patients with low L-carnitine levels, erythrocytes' viability is affected but the exact cause of this is unclear. It is hypothesized that there may be a disruption of membrane stability, which negatively affects erythrocytes' elasticity and osmotic resistance. It has been shown that dialysis patients with anemia have lower serum carnitine than non-anemic renal patients, and need higher doses of EPO [62].

Many studies have shown that L-carnitine favorably affects the impaired rheological properties of erythrocytes in hemodialysis patients [63–66]. L-carnitine affects the number and quality of erythrocytes through multiple modes of action. It is able to inhibit the apoptosis of progenitor cells, thereby increasing the number of red blood cells. It also improves the quality of erythrocytes by promoting phospholipid remodeling and improving viscoelastic properties [61].

The benefit of L-carnitine in dialysis-related anemia has been seen in numerous clinical studies with improvements in hematocrit levels, hemoglobin, and reduction of EPO dose [67–70].

A recent meta-analysis of 18 trials with 1090 participants examined the effects of L-carnitine on renal anemia in patients receiving hemodialysis [59]. The meta-analysis showed that L-carnitine therapy significantly:

- Increased plasma L-carnitine concentrations ($p < 0.00001$);
- Improved the response to erythropoiesis-stimulating agents (ESA) ($p < 0.00001$);
- Decreased the required ESA doses in patients receiving hemodialysis ($p < 0.00001$);
- Maintained hemoglobin and hematocrit levels.

In children, L-carnitine treatment is used only in those with ESA hyporesponsive anemia. A recent study showed an increase in hemoglobin levels and a decrease in ESA dose in children supplemented with oral L-carnitine 100 mg/kg per day in 3 doses up to 3 g/day for 6 months [71].

For hemodialysis-related anemia, the following L-carnitine administration dose has been suggested by various studies [72–75]:

- Adults: 20 mg/kg or 1–2 g L-carnitine IV after each dialysis session

- Children: 20 mg/kg after each dialysis session or
  - o      Children >40 kg: 1 g L-carnitine IV
  - o      Children 20–40 kg: 500 mg L-carnitine IV
  - o      Children 10–20 kg: 200 mg L-carnitine IV

Daily dialysis or daily hemodiafiltration used by some pediatric teams increases carnitine clearance [76]. In these situations, patients may need higher doses of L-carnitine.

### 5.2. L-Carnitine and Intradialytic Hypotension

Despite significant improvements in hemodialysis techniques in recent years, the frequency of recurrent intradialytic hypotension episodes has remained nearly unchanged with an incidence of 20–30% [77].

Although dialysis-related hypotension is frequently multifactorial, there are substantial data to indicate a significant element of cardiac dysfunction in hypotensive-prone dialysis patients [78]. Studies have shown an association between pre-dialysis hypotension and excess mortality, particularly for patients suffering from concomitant coronary disease, congestive heart failure, or a combination of the two [79–81].

In children, intradialytic hypotension episodes are more often related to inadequate dry weight. However, some pediatric patients continue to have intradialytic hypotensive episodes even after attempts to adapt dry weight accurately. These cases will ultimately require more frequent HD sessions [82–84].

Although the relationship between L-carnitine and the incidence of hypotensive episodes in hemodialysis patients has been widely investigated, the exact mechanism is not fully understood. The interaction between carnitine and the cardiovascular system is complex. L-carnitine has been shown to improve endothelial function [85,86], possibly since it normalizes nitric oxide (NO) production [87] and modulates endothelin-1 [88].

Hemodialysis patients with intradialytic hypotensive episodes have lower levels of total carnitine and free carnitine compared with hemodialysis patients without those episodes. This has been shown in adult and pediatric HD patients [44,89]. Several studies in adults have shown a significant reduction in the incidence of intradialytic hypotension with L-carnitine treatment [75,90,91].

For intradialytic hypotension recommended L-carnitine dose is:

- Adults: 20 mg/kg or 1–2 g L-carnitine IV after each dialysis session
- Children: 20 mg/kg or
  - o      Children >40 kg: 1 g L-carnitine IV
  - o      Children 20–40 kg: 500 mg L-carnitine IV
  - o      Children 10–20 kg: 200 mg L-carnitine IV

### 5.3. Role of Carnitine in Reducing Cardiovascular Complications

Patients with kidney disease have a high number of cardiovascular risk factors and are also exposed to uremia-specific risk factors that together increase cardiovascular mortality [92,93]. In addition, sudden cardiac death, resulting primarily from ventricular arrhythmias, accounts for the majority of cardiovascular deaths in patients with end-stage renal disease (ESRD), which appears to be unrelated to the presence of coronary artery disease [94].

L-carnitine plays a critical role in myocardial energy metabolism: it improves the utilization of fatty acids by transporting long-chain fatty acyl intermediates across the inner mitochondrial membrane for β oxidation. L-carnitine also regulates carbohydrate/glucose metabolism and reduces the toxic effects of long-chain acyl-CoA and acylcarnitine in myocytes.

L-carnitine treatment has been shown to be effective in increasing left ventricular ejection fraction, reducing arrhythmias such as ventricular ectopias, and decreasing left ventricular mass indexes in dialysis patients [95–98].

## 6. Future Perspectives in the Management of Kidney Disease: The Role of Genomics, Proteomics, Metabolomics, and the Microbiome

Next-generation DNA sequencing and new laboratory technologies have allowed researchers to begin analyzing the human genome, as well as the metabolome and proteome. The metabolome and the proteome are a collection of metabolites resulting from cellular metabolism and proteins expressed by activated genes, respectively. The metabolites and proteins are in a constant state of flux and at any given time reflect the status of cellular activity and health. The proteome and metabolome expression is unique for each body tissue—e.g., liver, muscle, kidney, etc.—and will change according to disease processes. It is not yet clear whether the change in proteome and metabolome causes the disease or if the disease is causing the changes.

The role of the gut and the gut microbiome should also be taken into account in the complex interplay with the body fluids, electrolytes, and uremic hemostasis [99,100]. In particular, the effects on the absorption of dietary nutrients, L-carnitine, and micronutrients will lead to influence at the metabolic level. Together, all these factors will play an important role in the overall body metabolism in kidney disease and, more importantly, impact further any underlying disease states, such as glucose tolerance, insulin resistance, and inflammatory and immune factors, which further influence kidney function [101].

Current research is focused on the analysis of metabolites, such as urea, creatinine, glucose, uric acid, and proteins (such as albumin, cystatin C, and complement) and parathyroid hormone to better understand kidney disease [102,103].

Further analysis of the complete spectrum of metabolites and proteins will not only allow the identification of biomarkers for diagnosis and treatment but also provide a better understanding of the underlying dysfunctional metabolic pathways that lead to kidney disease. The final goal should be to prevent or delay the onset of kidney disease by restoring the normal metabolome, proteome, and microbiome with multiple strategies including diet, supplementation with metabolic compounds such as L-carnitine, and possible gene therapies based on miRNA technologies [104–106].

The metabolome in relation to carnitine in patients with chronic kidney disease shows decreased carnitine palmitoyl transferase (CPT1) activity, a key enzyme in the carnitine cycle [107]. This reduced activity can lead to the accumulation of acylcarnitines and impairment of fatty acid oxidation in renal tubular epithelial cells in patients with kidney fibrosis. Restoring CPT1A activity has been shown to rebalance mitochondrial homeostasis in animal models [108]. Another recent study showed that restoring mitochondrial CPT1A activity protects against hypoxia and microtubular damage in the kidney [109].

L-carnitine together with a CPT1 stimulation significantly increased CPT1 activity and ATP levels and lowered renal malondialdehyde (MDA) and serum TNF-$\alpha$ concentration. This led to an improvement in renal histomorphology and a reduction in serum creatinine as well as blood urea nitrogen (BUN) [107].

Ischemic/reperfusion (I/R) injury can cause deleterious effects on kidney function by increasing oxidative stress and inflammatory biomarkers and decreasing L-carnitine levels in the kidneys. Studies have shown that L-carnitine supplementation can have nephroprotective effects on I/R injury by ameliorating all oxidative and inflammatory markers [108].

L-carnitine is a unique molecule essential for the metabolism of long-chain fatty acids s and the formation of ATP energy. L-carnitine administration in patients with kidney disease provides protective effects by improving metabolic balance, restoring the acetyl CoA pool, decreasing ROS levels, and improving mitochondrial metabolic flexibility. In addition, L-carnitine as an adjuvant therapy in dialysis patients can improve intradialytic complications, such as anemia hyporesponsive to EPO, intradialytic hypotension, muscle weakness, and reduced exercise capacity, leading to an improved quality of life.

**Funding:** This research received no external funding.

**Institutional Review Board Statement:** Not applicable.

**Informed Consent Statement:** Not applicable.

**Data Availability Statement:** Not applicable.

**Conflicts of Interest:** The authors declare no conflict of interest.

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
