# Peer review of "The Role of L-Carnitine in Kidney Disease and Related Metabolic Dysfunctions"

_kidneydial, doi:10.3390/kidneydial3020016_

Round 1

Reviewer 1 Report

Dear author, 

The issue is very interesting, and it focus on common supplement which is L-carnitine. The text need to some changing. I believe that if you do these, it can be more qualified. These are given below: 

1- Abbreviations should be given right places and text should be continue with these. (Ex. At Line 41 used β-oxidation but other lines used as beta oxidation)

2- Some of headings are related with L-carnitine but others are not. These should be standardised.

3- If it is possible, figure 2 can be reorganised. 

4-Some information about the L-carnitine could be given in the firstly or first paragraph can be written near the end of first part. 

5- Some connection sentences should be add the end of the parts.

Best regards

Author Response

We appreciate the feedback from the referees and hope to have made the revisions that are satisfactory and answer their valuable comments.

1- Abbreviations should be given right places and text should be continue with these. (Ex. At Line 41 used β-oxidation but other lines used as beta oxidation)

---- The abbreviations have been corrected in the manuscript (highlighted in yellow) as suggested by the referee.

2- Some of headings are related with L-carnitine but others are not.

These should be standardised.

---- We have changed 5. Role of L-carnitine in Dialysis related complications to 5. Dialysis related complications

3- If it is possible, figure 2 can be reorganised.

---- Figure 2 has been modified and simplified by removal of Ketogenesis part

4-Some information about the L-carnitine could be given in the firstly or first paragraph can be written near the end of first part.

Information on L-carnitine has been given in the first paragraph

---- Additional line has been added to the last paragraph as suggested - L-carnitine is a unique molecule essential for the metabolism long chain fatty acids across and formation of ATP energy

5- Some connection sentences should be added the end of the parts."

---- We have made the sentences more connected.

Reviewer 2 Report

The authors reviewed the role of L-carnitine deficiency in kidney disease especially AKI and dialysis-related complications. It would be better to include your search strategy and update your references. Please add Table(s) to represent the clinical trials with l-Carnitine supplementation and the outcome of patients with different kidney diseases. The article would require being revised significantly to improve the English. 

Author Response

Ref2

The authors reviewed the role of L-carnitine deficiency in kidney disease especially AKI and dialysis-related complications. It would be better to include your search strategy and update your references. Please add Table(s) to represent the clinical trials with l-Carnitine supplementation and the outcome of patients with different kidney diseases. The article would require being revised significantly to improve the English.

----  We have done a thorough search based on our own experience and knowledge and do not have a computer based strategy per se (as yet)

----  It's a good suggestion to add a Table, however since this review is mainly a general discussion maybe we will do that for a next review more focused on L-C and kidney disease outcomes.

---- The English has been improved (highlighted in yellow)

Reviewer 3 Report

The manuscript by Ulinski et al., titled: "The role of L-carnitine in kidney disease and related metabolic dysfunctions" is an interesting review on the topic of L-carnitine utilization and how it could impact outcome in kidney disease and related metabolic dysfunctions. 

The manuscript is well organized and logically structure, well-written and easy to follow for the reader. Good points are raised and it is a compact piece of work. One point from the reviewer alludes to the importance of the microbiome. The authors do not seem to have addressed that in their discussion. It would be interesting and a more comprehensive approach to include a paragraph discussing the involvement of the microbiome in regards to the kidney disease metabolic dysfunctions. Moreover the interplay of diabetes with kidney disease is particularly important and would warrant some mention and discussion. A paper that may be found useful in this regard is the following:

Sikalidis, A.K.; Maykish, A. The Gut Microbiome and Type 2 Diabetes Mellitus: Discussing A Complex Relationship. Biomedicines 2020, 8, 8. https://doi.org/10.3390/biomedicines8010008.

Good job overall.

Author Response

The manuscript by Ulinski et al., titled: "The role of L-carnitine in kidney disease and related metabolic dysfunctions" is an interesting review on the topic of L-carnitine utilization and how it could impact outcome in kidney disease and related metabolic dysfunctions.

The manuscript is well organized and logically structure, well-written and easy to follow for the reader. Good points are raised and it is a compact piece of work. One point from the reviewer alludes to the importance of the microbiome. The authors do not seem to have addressed that in their discussion. It would be interesting and a more comprehensive approach to include a paragraph discussing the involvement of the microbiome in regards to the kidney disease metabolic dysfunctions. Moreover the interplay of diabetes with kidney disease is particularly important and would warrant some mention and discussion. A paper that may be found useful in this regard is the following:

Sikalidis, A.K.; Maykish, A. The Gut Microbiome and Type 2 Diabetes Mellitus: Discussing A Complex Relationship. Biomedicines 2020, 8, 8. https://doi.org/10.3390/biomedicines8010008.

Good job overall.

---- We agree with the referee that the microbiome/diabetes is very important and we have added a paragraph page 9 as suggested (highlighted in yellow)

Reviewer 4 Report

This manuscript provides comprehensive review on the role of L-carnitine in kidney disease from the relationship of L-carnitine metabolism and kidney disease, an the use of L-carnitine in acute kidney injury and dialysis.  I have few suggestion to consider.

- Figure 2 was not referred in text 

- Some content are not pertinent (e.g., diet and chronic kidney disease, AKI from infection and drugs). In addition, several contents are redundant. would considerremoved to make review more concise and focused on L-carnitine. 

- The statement in page 9 line 344-345 are out of place, might be added in manuscript by error.

- consider add table to summarize related studies on the role of L-carnitine use in kidney disease and dialysis. This would make the data more accessible.

Author Response

Ref 4

This manuscript provides comprehensive review on the role of L-carnitine in kidney disease from the relationship of L-carnitine metabolism and kidney disease, an the use of L-carnitine in acute kidney injury and dialysis.  I have few suggestion to consider.

- Figure 2 was not referred in text

---- OK Referred to on page 3 line 98

- Some content are not pertinent (e.g., diet and chronic kidney disease, AKI from infection and drugs). In addition, several contents are redundant. would consider removing to make review more concise and focused on L-carnitine.

---- This is a good suggestion but we wanted to make this review a more general discussion (maybe next review)

- The statement in page 9 line 344-345 are out of place, might be added in manuscript by error.

---- Corrected, this line was there by error

- consider add table to summarize related studies on the role of L-carnitine use in kidney disease and dialysis. This would make the data more accessible.

---- Again this is a good suggestion but as before we wanted a more general review on this occasion.

Round 2

Reviewer 3 Report

The authors have made a reasonable effort in addressing reviewer's comments.